# The Effectiveness and Safety of Long-Term Macrolide Therapy for COPD in Stable Status: A Systematic Review and Meta-Analysis

**DOI:** 10.3390/diseases11040152

**Published:** 2023-10-27

**Authors:** Kazunori Nakamura, Yukio Fujita, Hao Chen, Kohei Somekawa, Fumihiro Kashizaki, Harumi Koizumi, Kenichi Takahashi, Nobuyuki Horita, Yu Hara, Shigeo Muro, Takeshi Kaneko

**Affiliations:** 1Department of Respiratory Medicine, Saiseikai Kumamoto Hospital, Kumamoto 861-4193, Japan; k.nakamura0826@gmail.com; 2Department of Respiratory Medicine, Nara Medical University, Nara 634-8522, Japan; ffyukio0314@gmail.com (Y.F.); smuro@naramed-u.ac.jp (S.M.); 3Department of Pulmonology, Yokohama City University Hospital, Yokohama 236-0004, Japan; somekawa51@yahoo.co.jp (K.S.); bronchiole0723@yahoo.co.jp (Y.H.); takeshi@yokohama-cu.ac.jp (T.K.); 4Department of Respiratory Medicine, Yokohama Minami Kyousai Hospital, Yokohama 236-0037, Japan; f.kashizaki@ymail.ne.jp (F.K.); h-koizumi@minamikyousai.jp (H.K.); ki-takahashi@minamikyousai.jp (K.T.); 5Chemotherapy Center, Yokohama City University Hospital, Yokohama 236-0004, Japan; horitano@yokohama-cu.ac.jp

**Keywords:** COPD, exacerbation, macrolide, azithromycin, erythromycin

## Abstract

**Background**: Chronic obstructive pulmonary disease (COPD) is a prevalent condition with fewer treatments available as the severity increases. Previous systematic reviews have demonstrated the benefits of long-term macrolide use. However, the therapeutic differences between different macrolides and the optimal duration of use remain unclear. **Methods**: A systematic review and meta-analysis were conducted to assess the effectiveness of long-term macrolide use in reducing COPD exacerbations, compare the therapeutic differences among macrolides, and determine the appropriate treatment duration. Four databases (PubMed, Cochrane Library, Web of Science, and ICHU-SHI) were searched until 20 March 2023, and a random-effects model was used to calculate the pooled effect. **Results**: The meta-analysis included nine randomized controlled trials involving 1965 patients. The analysis revealed an odds ratio (OR) of 0.34 (95% confidence interval [CI] 0.19, 0.59, *p* < 0.001) for the reduction in exacerbation frequency. Notably, only azithromycin or erythromycin showed suppression of COPD exacerbations. The ORs for reducing exacerbation frequency per year and preventing hospitalizations were −0.50 (95% CI: −0.81, −0.19; *p* = 0.001) and 0.60 (95% CI: 0.3, 0.97; *p* = 0.04), respectively. Statistical analyses showed no significant differences between three- and six-month macrolide prescriptions. However, studies involving a twelve-month prescription showed an OR of 0.27 (95% CI: 0.11, 0.68; *p* = 0.005; I^2^ = 81%). Although a significant improvement in St George’s Respiratory Questionnaire (SGRQ) total scores was observed with a mean difference of −4.42 (95% CI: −9.0, 0.16; *p* = 0.06; I^2^ = 94%), the minimal clinically important difference was not reached. While no adverse effects were observed between the two groups, several studies have reported an increase in bacterial resistance. **Conclusions**: Long-term use of azithromycin or erythromycin suppresses COPD exacerbations, and previous studies have supported the advantages of a 12-month macrolide prescription over a placebo.

## 1. Background

Chronic obstructive pulmonary disease (COPD) is a prevalent global ailment, with an estimated prevalence exceeding 10%. Data from the Global Burden of Disease Study 2017 indicated a COPD-attributed death rate of 42 per 100,000 (accounting for 4.72% of all-cause deaths) [1], furthermore making it one of the top three worldwide causes of death, with 90% of these fatalities occurring in low- and middle-income countries [2]. Individuals with COPD typically report dyspnea, activity limitations, and coughing with or without sputum production. They may also face acute respiratory events such as exacerbations, characterized by intensified respiratory symptoms that necessitate specific preventive and therapeutic interventions. COPD presents a significant public health challenge. It is not curable, but can be improved by adopting smoke-free lifestyles, minimizing exposure to air pollutants, and receiving vaccinations in a timely manner. Treatment options include medications, oxygen therapy, and pulmonary rehabilitation. Previous literature reviews have indicated that the economic impact on healthcare systems has escalated with the number of COPD exacerbations and the gravity of the disease [3]. The severity of COPD in elderly individuals is recognized as a risk factor for treatment failure during acute exacerbations, with recurring exacerbations heightening the risk of readmission [4,5]. Thus, reducing COPD exacerbations is imperative.

Pharmacological therapy for COPD aims to alleviate symptoms, decrease the frequency and severity of exacerbations, and enhance exercise tolerance and overall health. According to the GOLD 2023 guidelines, for patients whose condition worsens despite inhaled corticosteroids, long-acting β2-adrenergic agonists, and long-acting muscarinic antagonists, the addition of a macrolide, particularly azithromycin, is a consideration [6]. Studies have shown that azithromycin (250 mg/day or 500 mg three times per week) or erythromycin (250 mg two times per day) over a year can lower the risk of exacerbations in patients prone to acute exacerbations, when compared to standard care [7,8]. Non-pharmacological approaches including smoking cessation, pulmonary rehabilitation, and bronchoscopic interventions have also demonstrated efficacy in reducing mortality or preventing exacerbations. For individuals facing frequent exacerbations, interventions such as targeted lung denervation or bronchoscopic lung volume reduction could be considered to alleviate symptoms and ameliorate dyspnea. However, these interventions may be restricted due to factors such as pulmonary hypertension, frequent pneumonia, hypercapnia, and the absence of emphysematous COPD or collateral ventilation [9]. Therefore, less invasive medical interventions with fewer contraindications could offer a viable treatment path for COPD patients facing limited treatment options.

Macrolide drugs available on the market encompass immune-suppressive and antibiotic agents. Some of the latter have displayed anti-inflammatory effects, primarily due to their direct impact on inflammatory cells and processes during respiratory infections [10]. Recent research suggests that macrolide therapy represents an additional treatment option for moderate to severe COPD, particularly in preventing recurrent exacerbations that necessitate steroids and antibiotics [11]. Previous systematic reviews have also supported the utility of macrolides in preventing exacerbations and hospitalizations [12,13]. The frequency of exacerbations is closely linked to a rapid decline in lung function, compromised quality of life (QOL), and elevated mortality rates [14]. Given the ongoing progress in this field, further investigation is warranted to assess the impact of long-term macrolide therapy on the QOL of COPD patients. As a result, a systematic review was undertaken to investigate the potential advantages of sustained macrolide therapy in preventing exacerbations and hospitalizations while enhancing the QOL of individuals living with COPD.

## 2. Methods

### 2.1. Database Search Strategy

This study adhered to the guidelines of The Preferred Reporting Items for Systematic Reviews and Meta-Analyses (PRISMA) statement [15], and the study protocol was registered with the University Hospital Medical Information Network (study ID: UMIN000050691) [16]. A comprehensive search was conducted across four databases, namely PubMed, Cochrane Library, Web of Science, and ICHU-SHI, with data collection completed on 20 March 2023. Previous meta-analysis was conducted to identify the effect of macrolides, and the database search concluded in March 2019. The results of previous studies were used [12,17,18], and new studies in the database were searched from March 2019. The search strategy employed the following formula: (“Pulmonary Disease, Chronic Obstructive” OR “COPD” OR “Chronic obstructive pulmonary disease” OR “Emphysema” OR “Chronic bronchitis”) AND (“Azithromycin” OR “Erythromycin” OR “Clarithromycin” OR “Macrolide”).

Two review authors (K.N. and Y.F.) independently extracted relevant data regarding long-term macrolide therapy, the number of patients, and stable COPD. Initially, titles and abstracts were screened, followed by a thorough assessment of the full text. In cases where there were discrepancies in evaluations between the two reviewers, a third reviewer (H.C.) participated in the discussion.

### 2.2. Inclusion and Exclusion Criteria

This review was limited to articles published in English. The inclusion criteria for this review were as follows: (1) patients diagnosed with COPD; (2) a long duration of macrolide prescription in the trial group; (3) participants over 18 years old; and (4) macrolide use described during stable COPD status. The following exclusion criteria were applied: (1) studies that were not randomized controlled trials; and (2) studies from which data on the designated outcomes could not be extracted.

### 2.3. Definitions

The term “long-term duration” in this study refers to a period exceeding three months. The analysis in this study specifically focuses on three types of macrolides: azithromycin, erythromycin, and clarithromycin. This selection aims to accurately represent the conditions; only these three types of macrolides were available in Japan. As all the articles included in this study were published before 2023, the GOLD 2022 guideline was used to define moderate exacerbation, characterized by respiratory symptoms requiring additional treatment like antimicrobials or systemic steroids, without necessitating hospitalization. Severe exacerbation was defined as the occurrence of respiratory symptoms more severe than those observed in moderate exacerbation, resulting in hospitalization [19]. While the severity of COPD varied among the analyzed studies, the classification of COPD severity was based on the calculated post-bronchodilator value of forced expiratory volume in 1 s (FEV1) (% reference) in accordance with the GOLD guideline.

### 2.4. Outcomes

Both moderate and severe exacerbations of COPD were included in the analysis. The primary objective was to evaluate the effectiveness of interventions in preventing exacerbations of COPD. The secondary objectives included assessing the impact on hospitalization rates, the frequency of exacerbations per year, the St George’s Respiratory Questionnaire (SGRQ) scores [20], and the occurrence of adverse effects. Adverse effects were defined as the development of new symptoms that necessitated discontinuation of the prescribed treatment, regardless of their nature.

### 2.5. Statistics

The Endnote software, version 20.4 (Clarivate Analytics, London, UK), was utilized to manage the extracted data from the searched studies. The Review Manager software, version 5.3 (Cochrane Collaboration, Oxford, UK), was employed to conduct all the analyses in this study. Each outcome was estimated using the mean and standard error. To account for heterogeneity, a random-effects model was applied to calculate the pooled effect along with a 95% confidence interval (CI). The figures presented in the analysis were adjusted as necessary using the Review Manager software. Heterogeneity was assessed using I^2^ statistics, and the interpretation was as follows: I^2^ = 0%, indicating no heterogeneity; I^2^ > 0% but <25%, indicating minimal heterogeneity; I^2^ ≥ 25% but <50%, indicating mild heterogeneity; I^2^ ≥ 50% but <75%, indicating moderate heterogeneity; and I^2^ ≥ 75%, indicating strong heterogeneity [21]. A *p* value < 0.05 indicated statistical significance.

## 3. Results

### 3.1. Study Searching

The databases were searched from 1 March 2019, to 20 March 2023, yielding a total of 498 articles, with an additional manually searched study (Figure 1). Seventy-four articles were removed due to duplication, as identified by EndNote. In the initial screening, 286 articles were excluded either due to duplication or their lack of relevance to this review. Subsequently, 63 articles were eliminated during the second screening phase. Ultimately, after combining with previous studies, 11 articles were considered for data analysis; among these, 2 articles were excluded as their data could not be utilized. Consequently, a total of 9 articles were selected for the final analysis.

The characteristics of the included studies are presented in Table 1, encompassing references [7,8,22,23,24,25,26,27,28]. Among them, two studies were conducted in the UK and the Netherlands, while the rest were carried out in Australia, Belgium, China, Japan, and the USA. A total of 1965 patients were enrolled in these studies, with 976 patients undergoing macrolide therapy as part of this review. The FEV1 (% predicted) ranged from 37.2% to 53.7%, with most enrolled patients diagnosed with GOLD 3 severity of COPD. The duration of macrolide treatment varied across the studies: four studies adopted a 3-month regimen, one study opted for a 6-month course, and four studies employed a 12-month treatment period. All the included studies were randomized controlled trials (RCTs) that compared macrolide therapy to a placebo among patients with stable moderate or severe COPD. A predominant number of the studies demonstrated a low risk of bias, except for one study that indicated a high risk of performance bias and detection bias (Appendix A).

### 3.2. Preventing Exacerbations in COPD

Nine articles were analyzed to evaluate the prevention of exacerbation numbers compared to a placebo. The analysis showed that macrolides, when compared with placebo, had an odds ratio (OR) of 0.34 (95% CI: 0.19, 0.59, *p* < 0.001, I^2^ = 72%) quantified based on the number of patients with exacerbations (Figure 2). Subgroup analysis focusing on the differences between individual macrolide groups revealed significant variation. Both azithromycin and arythromycin were effective in inhibiting exacerbation, with ORs of 0.35 (95% CI: 0.16, 0.64, *p* < 0.991; I^2^ = 65%) and 0.20 (95% CI: 0.09, 0.45, *p* < 0.001; I^2^ = 40%), respectively. However, clarithromycin, according to one study, did not demonstrate effectiveness in preventing exacerbations.

Three studies were analyzed to evaluate the suppression of exacerbations per patient-year compared to a placebo. The meta-analysis revealed an odds ratio (OR) of 0.50 (95% CI: −0.81, −0.19; *p* = 0.001; I^2^ = 62%) (Appendix A). Subgroup analysis indicated a potential inclination towards the use of azithromycin in inhibiting exacerbations per patient-year, with an OR of −0.72 (95% CI: −1.61, 0.17, *p* = 0.12; I^2^ = 78%); however, no significant statistical difference was observed. On the other hand, an erythromycin study consisting of 31 patients demonstrated effectiveness, reporting an OR of −0.50 (95% CI: −0.84, −0.16; *p* = 0.004).

### 3.3. Optimal Prescription Duration

Four studies were conducted to assess the effectiveness of macrolides over a three-month period, and an additional four studies investigated their efficacy over a twelve-month period. Only one study specifically examined the effects of macrolide prescription after six months (Figure 3). No statistically significant differences were observed in the studies examining macrolide prescriptions at three months and six months. However, studies involving a twelve-month prescription showed an OR of 0.27 (95% CI: 0.11, 0.68; *p* = 0.005; I^2^ = 81%), indicating a significant reduction in the measured outcome. Subgroup analysis suggested a potential inclination towards the benefits of macrolide utilization, even within a three-month duration.

### 3.4. SGRQ

Three studies were included in the analysis to evaluate the effects of SGRQ compared to a placebo. The meta-analysis revealed a change from pretreatment of −4.42 (95% CI: −9.0, 0.16, *p* = 0.06, I^2^ = 94%) (Figure 4A). Based on empirical data and patient interviews, a mean change score of 4 units indicates a slightly effective treatment, 8 units for a moderately effective change, and 12 units for a highly effective treatment [29]. Despite the absence of a statistically significant difference between macrolide treatment and the placebo, the meta-analysis suggests a potential tendency towards the benefits of macrolide prescription in improving quality of life (QOL). Subgroup analysis was conducted based on SQRG symptom, impact, and activity scores, revealing a mean difference of −3.63 (95% CI: −5.73, 1.53; *p* < 0.01), −3.06 (95%CI: −8.26, 2.14; *p* = 0.25), and −2.24 (95% CI: 3.98, −0.51; *p* = 0.01), respectively (Appendix A).

### 3.5. Hospitalization Prevention

Four studies were included in the analysis about the prevention of hospitalization compared to a placebo. The meta-analysis revealed an OR of 0.60 (95% CI: 0.37–0.97; *p* = 0.04, I^2^ = 42%) (Figure 4B), indicating a significant effect. Subgroup analyses were conducted for macrolide groups and treatment duration, but no statistical differences were observed.

### 3.6. Adverse Events

Three studies were included in the analysis of adverse effects leading to discontinuation in comparison to a placebo. The meta-analysis revealed an OR of 1.29 (95% CI: 1.0, 1.66; *p* = 0.05; I^2^ = 0%) (Appendix A). Although there is a slight inclination towards a higher number of cases of medication continuation in the placebo group, it is not deemed clinically significant.

## 4. Discussion

This study presents compelling evidence that supports the long-term efficacy of using macrolides in patients with stable COPD. Macrolides effectively reduce the risk of exacerbations and hospitalizations without causing obvious adverse effects. Given the patient profile in the enrolled studies, a substantial portion experienced hospitalizations and frequent exacerbations per patient-year. Patients categorized under GOLD group E could potentially derive benefits from macrolide prescriptions. When comparing the duration of macrolide treatment for 3 or 6 months with a 12-month prescription, various treatment durations displayed a potential inclination towards the benefits of macrolide use. However, only the 12-month duration showed a statistically significant difference in preventing COPD exacerbations. Different macrolides exhibited varied effects in preventing COPD exacerbations. Azithromycin and erythromycin were identified as effective macrolides in averting COPD exacerbations, while there were insufficient data to substantiate the utility of clarithromycin in reducing COPD exacerbations. The impact of macrolides on improving the QOL for patients remained uncertain. Although statistically significant differences were observed in SGRQ scores compared to the placebo group, the minimal clinically important difference (MCID) was not achieved.

COPD is characterized by an increased presence of macrophages in the outer airways, lung tissue, and pulmonary blood vessels. Additionally, there are higher numbers of activated neutrophils and lymphocytes [30]. These inflammatory cells, alongside epithelial cells, and other structural components, release various inflammatory substances. The inflammation observed in the lungs of patients with COPD seems to be a modified version of the typical inflammatory response to persistent irritants, such as cigarette smoke [31]. Three studies identified the ratio of current smokers, ranging from 21% to 45% [8,25,26]. However, there were no subgroup analyses comparing the different effects of macrolides in former and current smokers. The different effects of macrolides in current and former smokers remain unclear. Lung inflammation can continue to persist even after individuals have stopped smoking, and the precise reasons for this phenomenon remain unclear. The specific mechanisms underlying this heightened inflammation have not been comprehensively elucidated. Network analysis and GWAS studies have identified multiple mechanisms of intracellular processes and genetic factors associated with COPD [32,33].

Exacerbations of respiratory symptoms in individuals with COPD can arise from numerous factors, either independently or in combination. These factors encompass respiratory infections triggered by bacteria or viruses, which might concurrently exist, as well as environmental pollutants or unidentified triggers [34]. During these exacerbation episodes, there is observable evidence of heightened inflammation within the airways and throughout the body. This escalation in inflammation leads to a greater retention of gases and an excessive inflation of the lungs, consequently resulting in diminished airflow during exhalation. Currently, it is understood that specific macrolide antibiotics exhibit multiple effects on inflammation. These effects encompass direct modulatory actions, both inhibitory and stimulatory, on leukocytes over time. These effects are attributed to their persistent intracellular accumulation, as particularly seen with azithromycin, leading to inhibitory effects on neutrophils, eosinophils, and macrophages [10]. Elevated levels of neutrophil-induced inflammation are associated with an increased risk of COPD exacerbations [35]. Macrolides encompass an array of direct effects on neutrophil function, spanning from an acute enhancement of degranulation and the release of reactive oxygen species (ROS) and certain cytokines, to a subsequent inhibition of neutrophil infiltration and the release of mediators [36].

Given that viral infections play a significant role as underlying factors in some instances of exacerbations in COPD, it is likely that the positive outcomes observed with the use of azithromycin in these patients could also be attributed to the direct antiviral effects of this macrolide antibiotic [37]. In vitro studies have demonstrated that azithromycin, unlike erythromycin or telithromycin, has the capability to enhance the expression of interferons induced by rhinovirus and promote the production of corresponding proteins [38]. Furthermore, it has been shown to diminish the replication of rhinovirus in bronchial epithelial cells. Beyond this, azithromycin has shown potential in inhibiting the replication of various viruses in vitro, including but not limited to influenza A, Zika, Ebola, coronaviruses, and enteroviruses. Notably, it was also associated with a decrease in viral load among children in Niger infected with α- or β-coronaviruses [37]. Moreover, azithromycin exhibited the ability to impede endocytosis by host cells of newly formed viral progeny, a property not shared by other anti-influenza medications.

In the realm of respiratory disorders, it is crucial to consider azithromycin’s ability to hinder mucus secretion. The excessive production of mucus is a hallmark of various inflammatory airway conditions. In COPD patients, increased mucus secretion is linked to a negative impact on their QOL [39]. The severity of phlegm production and productive cough is associated with more frequent COPD exacerbations, increased breathlessness, and a further decline in patient-reported QOL [40]. Azithromycin’s potential to reduce the expression and release of mucin 5AC (MUC5AC) from airway epithelial goblet cells, triggered by lipopolysaccharides (LPS) or tumor necrosis factor alpha (TNFα), has been recognized [41]. The inhibition of MUC5AC gene expression involves the suppression of ERK1/2 and JNK phosphorylation, leading to a decrease in the activities of NFκB and AP-1 pathways. These pathways are also involved in the secretion of MMP-9 and IL-8 [42].

In this meta-analysis, both azithromycin and erythromycin demonstrated the ability to control the exacerbation of COPD in subgroup analysis. Macrolides possess not only antimicrobial activity but also other beneficial functions for COPD, such as immunomodulation, inhibition of mucous secretion, and enhancement of airway barrier functions [36]. In fact, the use of macrolides has been based more on their potential anti-inflammatory and immunoregulatory effects, which have been demonstrated in both in vitro experimental models and other inflammatory lung diseases. As immunomodulators, they play a role in regulating cytokines such as interleukin (IL)-Ⅰβ, IL-6, IL-10, and TNF-α [43]. Particularly, erythromycin and clarithromycin have been shown to suppress the production of IL-8 and IL-33 during RS virus infection [43]. Azithromycin inhibits the production of proinflammatory cytokines, reduces neutrophil infiltration, and promotes macrophage accumulation [44]. Previous studies have indicated that the number of neutrophils and cytokines, such as TNF-α and IL-8, increase during COPD exacerbation [45]. A crucial macrophage-mediated action is the ability of macrolides to promote macrophage phenotype conversion from M1 to M2-like macrophages, which generate inflammation-dampening mediators, such as IL-10, and exhibit enhanced phagocytic capacity, thereby promoting resolution of inflammation [46]. Macrolides may contribute to the inhibition of exacerbation by suppressing these cytokines and impeding neutrophil migration.

It is possible that systemic inflammation is present, which could contribute to the additional health conditions often observed in COPD patients. Dysregulation of the immune system at this critical site is associated with a range of chronic inflammatory COPD conditions [47]. In patients with COPD who undergo frequent exacerbations, the available treatment options are restricted due to low pulmonary function, intolerance to invasive interventions, and other factors. Preventing exacerbations is a crucial consideration in managing the condition. As per the GOLD 2023 guidelines, one potential approach is the utilization of macrolides for patients who continue to experience exacerbations despite receiving appropriate treatment [6]. However, there is insufficient robust evidence to support this recommendation, and the optimal duration of macrolide treatment remains unclear. Our study indicated that a 12-month macrolide course effectively suppressed COPD exacerbations. Unfortunately, there were limited randomized controlled trials exploring the usage of macrolides for more than 12 months in COPD patients. A recent retrospective cohort study, following the COLUMBUS study, found no statistically significant difference between the group that received macrolides for 24 months and the group that received macrolides for 12 months before discontinuation [48]. Nevertheless, other retrospective studies have shown that the use of azithromycin 500 mg three times a week for 36 months reduced hospitalizations by 39% in COPD patients experiencing frequent exacerbations [49]. Further randomized controlled trials investigating the use of macrolides beyond one year are necessary.

The macrolide group demonstrated statistically significant enhancements in total, symptom, and activity scores as evaluated through the SGRQ assessment. The SGRQ score serves as a valuable tool for the assessment of chronic airflow limitation [50], representing one of the most extensively documented comprehensive measures. Scores below 25 are infrequent among diagnosed COPD patients, while scores equal to or greater than 25 are highly unusual in healthy individuals [51,52]. Previous systematic reviews have indicated a correlation between improvements in the SGRQ score and a decreased risk of COPD exacerbations [53]. A recent study brought to light that COPD patients participating in the STATCOPE or MACRO studies but not receiving macrolide prescriptions experienced a short-term decline in symptom scores that were severely impaired [54,55]. Nonetheless, the implementation of long-term macrolide therapy did not exhibit a noteworthy enhancement in the SGRQ score beyond the MCID. Further research is imperative to explore the potential benefits of macrolide usage among individuals with COPD who grapple with inadequately managed symptoms.

While macrolide therapy undoubtedly offers benefits in reducing exacerbations of COPD and asthma, these advantages must be carefully considered, given the potential adverse effects, which include the development of antimicrobial resistance. Several clinical trials have examined the impact of macrolides on antibiotic resistance in patients with COPD. One study, called the MACRO study, involved the collection of nasopharyngeal swabs. Among participants who initially did not carry specific respiratory pathogens but later became colonized during the study, cultures from 68% of the individuals in the azithromycin group and 70% in the placebo group were available for susceptibility testing. The incidence of macrolide resistance was discovered to be 81% in the azithromycin group and 41% in the placebo group (*p* < 0.001) [8]. In the AZISAST study, a subgroup underwent oropharyngeal swab testing to evaluate macrolide resistance. At the study’s commencement, 47.8% of patients on azithromycin and 39.1% on placebo were colonized with erythromycin-resistant streptococci. By the conclusion of the treatment period, the proportions of erythromycin-resistant streptococci had risen to 87% among those taking azithromycin and 35% among those taking placebo [28]. Within a subset of the AMAZES trial, where paired sputum samples were examined from 34 patients on placebo and 27 on azithromycin, significant increases were noted in five macrolide resistance genes and two tetracycline resistance genes among patients treated with azithromycin [56]. The long-term use of macrolides is recommended by the British Thoracic Society and GOLD for COPD patients. However, there are limited data verifying the effectiveness of long-term macrolides in real-world settings. Despite the concerning development of antimicrobial resistance, constraints within the insurance system may also hinder the widespread use of macrolides.

### Limitation

This study has several limitations. Firstly, moderate-to-strong heterogeneity was observed in this meta-analysis, which may undermine the robustness of the conclusion. Secondly, bacterial outcomes, such as antibiotic resistance analysis, were not conducted due to variations in the results’ presentation across different papers, and the wide range of detected bacteria and standardized outcomes not being available. Thirdly, only one study identified the effects of clarithromycin, which may render the conclusion regarding clarithromycin not sufficiently reliable. Fourthly, it is worth noting that other variables, such race and blood eosinophil counts, may potentially affect the efficacy of macrolide therapy in long-term use for COPD patients. However, there is a scarcity of data from RCTs that can robustly support a comprehensive subgroup analysis.

## 5. Conclusions

Long-term use of azithromycin or erythromycin suppresses exacerbation of COPD, and previous studies have supported the superiority of a 12-month macrolide prescription over a placebo.

## Figures and Tables

**Figure 1 diseases-11-00152-f001:**
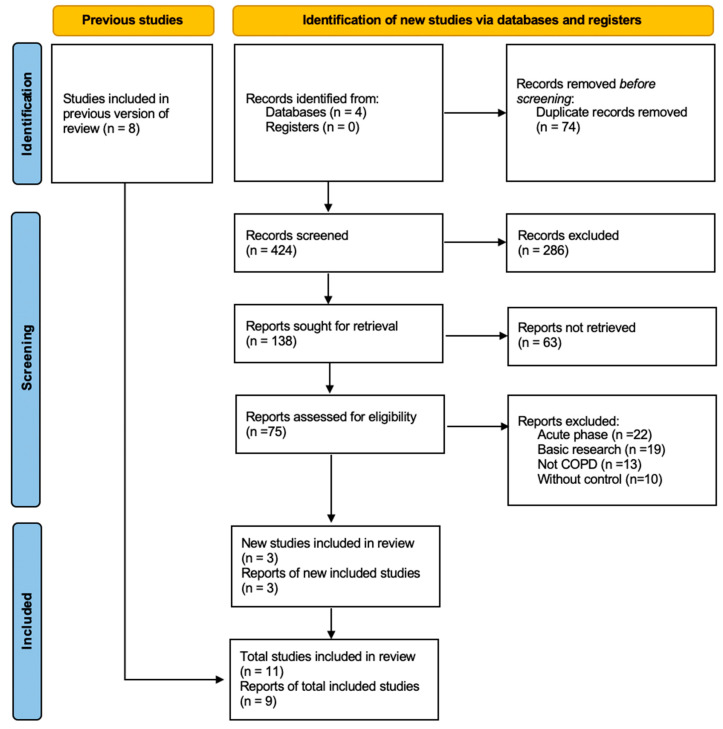
PRISMA flowchart diagram.

**Figure 2 diseases-11-00152-f002:**
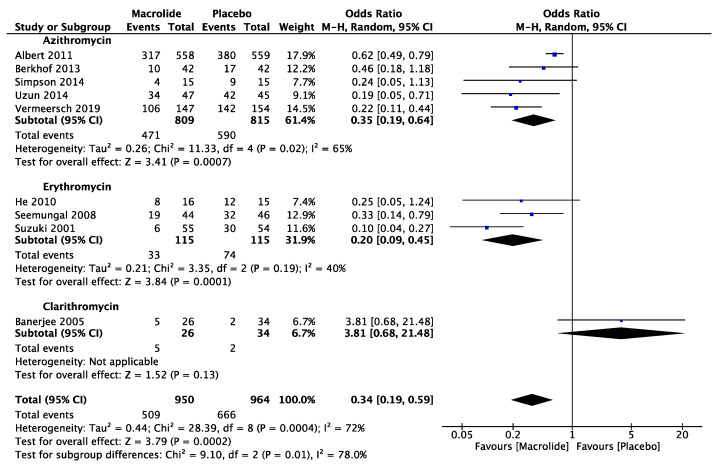
The effect of different macrolides in prevention of exacerbation [7,8,22,23,24,25,26,27,28].

**Figure 3 diseases-11-00152-f003:**
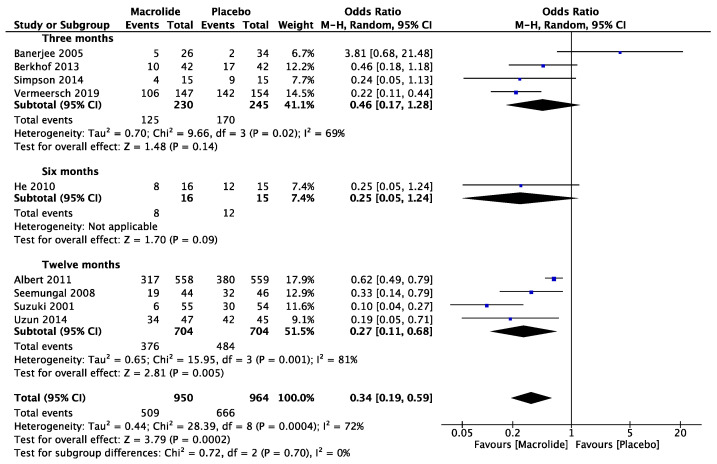
The effect of macrolides in prevention of exacerbation by different duration [7,8,22,23,24,25,26,27,28].

**Figure 4 diseases-11-00152-f004:**
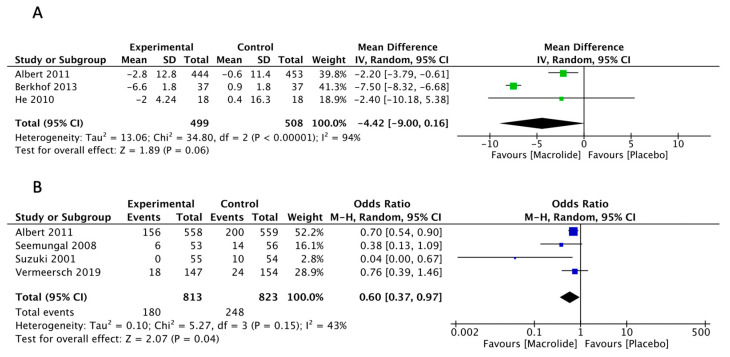
(**A**). The effect of macrolides in reducing St George’s Respiratory Questionnaire total scores [8,22,24]. (**B**). The effect of macrolides in hospitalization prevention [7,8,27,28].

**Table 1 diseases-11-00152-t001:** Characteristics of included studies.

Author and Year	Country	Design	Medicine	Usage	Severity	Treatment	Control	Duration
Albert 2011 [8]	USA	RCT	Azithromycin	250 mg once daily	≥moderate	570	572	12 months
Banerjee 2005 [26]	UK	RCT	Clarithromycin	500 mg once daily	≥moderate	31	36	3 months
Berkhof 2013 [22]	Netherlands	RCT	Azithromycin	250 mg once 3 days/week	≥moderate	42	42	3 months
He 2010 [24]	China	RCT	Erythromycin	125 mg 3 times daily	≥moderate	16	15	6 months
Seemungal 2008 [7]	UK	RCT	Erythromycin	250 mg twice daily	≥moderate	53	56	12 months
Simpson 2014 [25]	Australia	RCT	Azithromycin	250 mg once daily	≥moderate	15	15	3 months
Suzuki 2001 [27]	Japan	RCT	Erythromycin	200–400 mg once daily	≥moderate	55	54	12 months
Uzun 2014 [23] ^†^	Netherlands	RCT	Azithromycin	500 mg once 3 days/week	≥moderate	47	45	12 months
Vermeersch 2019 [28]	Belgium	RCT	Azithromycin	250 mg every 2 days	≥moderate	147	154	3 months

RCT: randomized controlled trial; †: three cases of mild severity.

## Data Availability

Data is available from the corresponding authors on request.

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
