# Peer review of "The Effectiveness and Safety of Long-Term Macrolide Therapy for COPD in Stable Status: A Systematic Review and Meta-Analysis"

_diseases, 2023, doi:10.3390/diseases11040152_

Round 1
Reviewer 1 Report
Comments and Suggestions for Authors
The article is dedicated to the effectiveness and safety of long-term macrolide therapy for COPD in stable status. The topic discussed is crucial for the prevention of exacerbations of respiratory infections.
I would like to make a few comments:
1) Lines 52-53: “COPD….. , being both preventable and treatable.”
Comment: “COPD isn’t curable, but it can get better by not smoking, avoiding air pollution and getting vaccines. It can be treated with medicines, oxygen and pulmonary rehabilitation” …. (https://www.who.int/news-room/fact-sheets/detail/chronic-obstructive-pulmonary-disease-(copd ) )
2) Line 53: “Despite the existence of effective treatments...”
Comment: There is no cure for COPD. Treatment can only relieve symptoms and increase the duration of the periods of remission.
3) It is better to remove “Figure1. PRISMA flowchart diagram” from supplementary material to the main text.
4) Line 148: “The database was searched...”
Comment: You used four databases. Correct please.
5) In the Discussion
Lines 255-267: The specific mechanisms underlying this heightened inflammation have not been comprehensively elucidated...
Comment: It is need to correct. Authors should read and cite articles that, based on network analysis and GWAS studies, have identified multiple mechanisms of intracellular processes and genetic factors associated with COPD:
Hoeng, J. Boue, S. Fields, B. Park, J. Peitsch, M.C. Schlage, W.K. Talikka, M. Binenbaum, I. Bondarenko, V. Bulgakov, O.V. Cherkasova, V. Diaz-Diaz, N., et al. Enhancement of COPD biological networksusing a web-based collaboration interface. F1000Research. 2015;4.
Silverman EK. Genetics of COPD. Annu Rev Physiol. 2020 Feb 10;82:413-431. doi: 10.1146/annurev-physiol-021317-121224.
6) Were antimycotics prescribed in the described studies to prevent fungal diseases when antibiotic therapy was applied? If yes, then it should be mentioned.
Author Response
Reviewer 1
Comment 1: Lines 52-53: “COPD..., being both preventable and treatable.” “COPD isn’t curable, but it can get better by not smoking, avoiding air pollution and getting vaccines. It can be treated with medicines, oxygen and pulmonary rehabilitation” (https://www.who.int/news-room/fact-sheets/detail/chronic-obstructive-pulmonary-disease-(copd ) )
Response: We appreciate the reviewer's comments on this point. Our expression is not correct. We modified our manuscript as follows: “COPD presents a significant public health challenge. It is not curable, but it can improve by avoiding smoking, minimizing exposure to air pollution, and receiving vaccinations.” (Line 53-55)
Comment 2: Line 53: “Despite the existence of effective treatments...” There is no cure for COPD. Treatment can only relieve symptoms and increase the duration of the periods of remission.
Response: We appreciate the reviewer's comments on this point. Our expression is not correct. We modified our manuscript as follows “It is not curable, but it can improve by avoiding smoking, minimizing exposure to air pollution, and receiving vaccinations. Treatment options include medications, oxygen therapy, and pulmonary rehabilitation.” (Line 54-56)
Comment 3: It is better to remove “Figure1. PRISMA flowchart diagram” from supplementary material to the main text.
Response: We appreciate the reviewer's comments on this point. We have moved the PRISMA flowchart diagram from supplementary material to the main test. (Line 166-168)
Comment 4: Line 148: “The database was searched...” You used four databases. Correct please.
Response: We appreciate the reviewer's comments on this point. Our expression is not correct. We corrected our manuscript as follows “The databases were searched from March 1, 2019, to March 20, 2023, yielding a total of 498 articles, with an additional manually searched study” (Line 159-160)
Comment 5: In the Discussion Lines 255-267: The specific mechanisms underlying this heightened inflammation have not been comprehensively elucidated... It is need to correct. Authors should read and cite articles that, based on network analysis and GWAS studies, have identified multiple mechanisms of intracellular processes and genetic factors associated with COPD:
Response: We appreciate the reviewer's comments on this point. Our expression is not correct. We have modified our manuscript and cited the relevant references as follows: “The specific mechanisms underlying this heightened inflammation have not been comprehensively elucidated. Network analysis and GWAS studies have identified multiple mechanisms of intracellular processes and genetic factors associated with COPD [32,33].” (Line 268-271)
Comment 6: Were antimycotics prescribed in the described studies to prevent fungal diseases when antibiotic therapy was applied? If yes, then it should be mentioned.
Response: We appreciate the reviewer's comments on this point. We conducted a search on the prophylactic administration of antimycotics in COPD; however, there were limited studies discussing this topic.
Reviewer 2 Report
Comments and Suggestions for Authors
The manuscript, entitled “The effectiveness and safety of long-term macrolide therapy for COPD in stable status: a systemic review and meta-analysis” described that to survey the treatment effect of macrolides on Chronic Obstructive Pulmonary Disease, the authors conducted a systematic review and meta-analysis and found that Long-term use of azithromycin or erythromycin suppresses COPD exacerbations. However, due to some significant drawbacks, my suggestion is rejection.
Comments:
1. The article surveys the short time span of published articles on COPD, screening only 498 articles published over a four-year period from 1 March 2019 to 20 March 2023. However, there are certainly many eligible articles that were published before 1 March 2019. Why did the authors screen only articles from this time period for their study?
2. The sample size is too small. In line 154, it is stated that a total of 9 articles were considered for data analysis, which were picked from 498 articles. However, for a systematic review and meta-analysis paper, the sample size of 9 is too small which can lead to less comprehensive results. What’s more, the authors clarified that the analysis in this study specifically focused on three types of macrolides: azithromycin, erythromycin, and clarithromycin, but from table 1 we can see that only on paper is about clarithromycin. So, the conclusion that Clarithromycin did not demonstrate effectiveness in preventing exacerbation might not accurate.
3. The survey did not draw any meaningful conclusions. The conclusion the authors get from this paper is that long-term use of azithromycin or erythromycin suppresses COPD exacerbations, and previous studies have supported the superiority of a 12-month macrolide prescription over a placebo. But the conclusion is not novelty, and in the results section the authors did not analyze the factor of duration time.
4. The format of figures and tables is unsuitable. For figure 1, the figure legend should be put below the figure and for table 1, the line above the table name should be deleted.
5. The English writing should be polished by native English speaker, as there are some grammar errors in the text. For example, what did the authors try to express by the sentence “the differences between macrolides and optimal duration are still unclear.”?
Comments on the Quality of English LanguageThe English writing should be polished by native English speaker, as there are some grammar errors in the text. For example, what did the authors try to express by the sentence “the differences between macrolides and optimal duration are still unclear.”?
Author Response
Dear Reviewer
We appreciate your comments aimed at improving the quality of this manuscript.
- The article surveys the short time span of published articles on COPD, screening only 498 articles published over a four-year period from 1 March 2019 to 20 March 2023. However, there are certainly many eligible articles that were published before 1 March 2019. Why did the authors screen only articles from this time period for their study?
Response: We appreciate the reviewer's comments on this point. Our expression is not clear here. There were three meta-analyses or reviews published around 2019 that discussed the effect of long-term macrolide use on stable COPD (PMID: 31564856; PMID: 34058115; PMID: 33448349). The overlapping date of the database searches in these previous studies concluded in June 2019. We combined the results of these previous studies and conducted the meta-analysis from March 2019. We added this information in the manuscript as follows “Previous meta-analysis was conducted to identify the effect of macrolides, and the database search concluded in March 2019. The results of previous studies were used [12, 17, 18], and new studies in the database were searched from March 2019.” (Line 101 -103)
- The sample size is too small. In line 154, it is stated that a total of 9 articles were considered for data analysis, which were picked from 498 articles. However, for a systematic review and meta-analysis paper, the sample size of 9 is too small which can lead to less comprehensive results. What’s more, the authors clarified that the analysis in this study specifically focused on three types of macrolides: azithromycin, erythromycin, and clarithromycin, but from table 1 we can see that only one paper is about clarithromycin. So, the conclusion that Clarithromycin did not demonstrate effectiveness in preventing exacerbation might not accurate.
Response: We appreciate the reviewer's comment on this point. We made every effort to conduct a comprehensive database search, which included one comprehensive database covering conference reports in Japan. However, as the reviewer pointed out, there were only nine RCT studies included in this meta-analysis, with only one RCT study examining the effect of clarithromycin. We have added an explanation of the limitations in the limitation section: “Thirdly, the limited availability of studies examining the effects of clarithromycin restricts the reliability of conclusions related to clarithromycin.” (Line 393-395)
- The survey did not draw any meaningful conclusions. The conclusion the authors get from this paper is that long-term use of azithromycin or erythromycin suppresses COPD exacerbations, and previous studies have supported the superiority of a 12-month macrolide prescription over a placebo. But the conclusion is not novelty, and in the results section the authors did not analyze the factor of duration time.
Response: We appreciated the reviewers comment on this point. We added the sub-analysis of between different duration and the effect of macrolides as follows “Four studies were conducted to assess the effectiveness of macrolides over a three-month period, and an additional four studies investigated their efficacy over a twelve-month period. Only one study specifically examined the effects of macrolide prescription after six months (Figure 3). No statistically significant differences were observed in the studies examining macrolide prescriptions at three months and six months. However, studies involving a twelve-month prescription showed an OR of 0.27 (95% CI: 0.11, 0.68; p=0.005; I2 = 81%), indicating a significant reduction in the measured outcome. Subgroup analysis suggested a potential inclination towards the benefits of macrolide utilization, even within a three-month duration.” (Line 204 - 212)
- The format of figures and tables is unsuitable. For figure 1, the figure legend should be put below the figure and for table 1, the line above the table name should be deleted.
Response: We appreciated the reviewers comment on this point. We have modified our figure legends and deleted the line above the table name. (Line 181)
- The English writing should be polished by native English speaker, as there are some grammar errors in the text. For example, what did the authors try to express by the sentence “the differences between macrolides and optimal duration are still unclear.”
Response: We appreciated the reviewers comment on this point. We have asked Three native English speakers helped us to improve the quality of this manuscript. (Line 405, 406)
Reviewer 3 Report
Comments and Suggestions for Authors
The paper "The effectiveness and safety of long-term macrolide therapy for 2 COPD in stable status: a systemic review and meta-analysis" is well written and interesting. However, I think that authors could improve the paper and mainly the discussion section (and possibly the results section if data are available). As authors are expert into the argument, I would be very happy if they attempted to explain why are studies on macrolide use in stable COPD not recent, and, despite the good results, the use of macrolide is currently not very common. I suggest that authors add limitation of macrolide use into the abstract section (i.e., adverse effects and increse in bacterial resistance rate). May authors add some real world data on current macrolide use in COPD? As authors know, recent studies have raised the interest on blood eosinophil counts in COPD. It would be interesting if they specified the link between macroilide use and eosinophils. Did they feel that subjects with low eosinophil counts (<100/mcl) is a target for macrolide use? In authors'opinion which are the Group E GOLD COPD? Which is the link between former and current smokers and macrolide use? Did they think that racial reasons can modify the significance (effectiveness and adverse effects) of macrolide in COPD? May authors add some data on maintenance inhaler treatment for subjects enrolled in RCTs on macrolide use? If authors can introduce these data, the review would be more interesting
Author Response
Reviewer 3
Comment 1: However, I think that authors could improve the paper and mainly the discussion section (and possibly the results section if data are available). As authors are expert into the argument, I would be very happy if they attempted to explain why are studies on macrolide use in stable COPD not recent, and, despite the good results, the use of macrolide is currently not very common.
Response: We appreciated the reviewers comment on this point. We have added information to our manuscript as follows: "In fact, the use of macrolides has been based more on their potential anti-inflammatory and immunoregulatory effects, which have been demonstrated in both in vitro experimental models and other inflammatory lung diseases." "Despite the concerning development of antimicrobial resistance, the insurance system may also inhibit the widespread use of macrolides."(Line 317-319, Line 385-387)
Comment 2: I suggest that authors add limitation of macrolide use into the abstract section (i.e., adverse effects and increase in bacterial resistance rate). May authors add some real-world data on current macrolide use in COPD?
Response: We appreciated the reviewers comment on this point. We added this information in abstract section "Although no adverse effects were observed between the two groups, several studies have reported an increase in bacterial resistance.” We also added this information in discussion: “long-term use of macrolides is recommended by the British Thoracic Society and GOLD for COPD patients. However, there is limited data verifying the effectiveness of long-term macrolides in real-world settings.” (Line 382-385)
Comment 3: As authors know, recent studies have raised the interest on blood eosinophil counts in COPD. It would be interesting if they specified the link between macroilide use and eosinophils. Did they feel that subjects with low eosinophil counts (<100/mcl) is a target for macrolide use?
Response: We appreciate the reviewer's comment on this point. We searched the database, and there were limited studies identifying the effect of macrolides in COPD with low eosinophil counts (<100/mcl). On the other hand, several studies identified an effect of macrolides compared with placebo on the rate of exacerbations requiring hospitalization (PMID: 34807989). Since this meta-analysis focused on COPD, we didn't add information about the effect of macrolides in asthma.
Comment 4: In authors' opinion which are the Group E GOLD COPD? Which is the link between former and current smokers and macrolide use?
Response: We appreciated the reviewers comment on this point. Group E COPD includes patients with high risk (≥2 exacerbations per year or ≥1 requiring hospitalization) and any level of symptoms, according to GOLD 2023. It combines Group C and D from the previous version. Three studies identified the ratio of current smokers, verified to range from 21% to 45%. However, there were no subgroup analyses comparing the different effects of macrolides in former and current smokers. The different effects of macrolides in current and former smokers remain unclear. We added this information in our manuscript as follows: “Three studies identified the ratio of current smokers, verified to range from 21% to 45% [8,25,26]. However, there were no subgroup analyses comparing the different effects of macrolides in former and current smokers. The different effects of macrolides in current and former smokers remain unclear.” (Line 264-267)
Comment 5: Did they think that racial reasons can modify the significance (effectiveness and adverse effects) of macrolide in COPD?
Response: We appreciate the reviewer's comment on this point. Albert et al. conducted an RCT containing 1142 patients, of which 82% were White, while He et al. and Suzuki et al. conducted two studies in Asian populations. There was no data to support different effectiveness and adverse effects in particular racial populations.
Comment 6: May authors add some data on maintenance inhaler treatment for subjects enrolled in RCTs on macrolide use? If authors can introduce these data, the review would be more interesting
Response: we appreciate the reviewer's comment on this point. There were no studies that described the data for maintenance inhaler treatment, making it impossible to confirm changes in the inhaled drugs for each patient during the follow-up period.
Round 2
Reviewer 2 Report
Comments and Suggestions for Authors
Comments:
1. First of all the most serious flaw of the study, the problem of too small a sample size was not modified. The sample size is too small. In line 166, it is stated that a total of 9 articles were considered for data analysis, which were picked from 498 articles. However, for a systematic review and meta-analysis paper, the sample size of 9 is too small which can lead to less comprehensive results. What’s more, the authors clarified that the analysis in this study specifically focused on three types of macrolides: azithromycin, erythromycin, and clarithromycin, but from table 1 we can see that only on paper is about clarithromycin. So, the conclusion that Clarithromycin did not demonstrate effectiveness in preventing exacerbation might not accurate.
2. The survey did not draw any meaningful conclusions. The conclusion the authors get from this paper is that long-term use of azithromycin or erythromycin suppresses COPD exacerbations, and previous studies have supported the superiority of a 12-month macrolide prescription over a placebo. But the conclusion is not novelty, and in the results section the authors did not analyze the factor of duration time.
3. In addition to the type of medication used, which can have an impact on the condition, other factors, including the patient's underlying condition, can have an impact on the outcome, and the authors did not consider factors other than medication use.
4.The formatting of references is not consistent.
Comments on the Quality of English LanguageQuality of English writing could be improved
Author Response
Dear Reviewer
We appreciate the reviewers comment in improving the quality of this manuscript.
Comment 1. First of all the most serious flaw of the study, the problem of too small a sample size was not modified. The sample size is too small. In line 166, it is stated that a total of 9 articles were considered for data analysis, which were picked from 498 articles. However, for a systematic review and meta-analysis paper, the sample size of 9 is too small which can lead to less comprehensive results. What’s more, the authors clarified that the analysis in this study specifically focused on three types of macrolides: azithromycin, erythromycin, and clarithromycin, but from table 1 we can see that only one paper is about clarithromycin. So, the conclusion that Clarithromycin did not demonstrate effectiveness in preventing exacerbation might not accurate.
Response: We appreciated the reviewer’s comment one this point. We made every effort to conduct a comprehensive database search, which included one comprehensive database covering conference reports in Japan. However, as the reviewer pointed out, there were only nine RCT studies included in this meta-analysis, with only one RCT study examining the effect of clarithromycin. We have added an explanation of the limitations in the limitation section: “Thirdly, the limited availability of studies examining the effects of clarithromycin restricts the reliability of conclusions related to clarithromycin.” (Line 393-395)
Comment 2. The survey did not draw any meaningful conclusions. The conclusion the authors get from this paper is that long-term use of azithromycin or erythromycin suppresses COPD exacerbations, and previous studies have supported the superiority of a 12-month macrolide prescription over a placebo. But the conclusion is not novelty, and in the results section the authors did not analyze the factor of duration time.
Response: We appreciated the reviewers comment on this point. We added the sub-analysis of between different duration and the effect of macrolides as follows “Four studies were conducted to assess the effectiveness of macrolides over a three-month period, and an additional four studies investigated their efficacy over a twelve-month period. Only one study specifically examined the effects of macrolide prescription after six months (Figure 3). No statistically significant differences were observed in the studies examining macrolide prescriptions at three months and six months. However, studies involving a twelve-month prescription showed an OR of 0.27 (95% CI: 0.11, 0.68; p=0.005; I2 = 81%), indicating a significant reduction in the measured outcome. Subgroup analysis suggested a potential inclination towards the benefits of macrolide utilization, even within a three-month duration.” (Line 204 - 212)
Comment 3. In addition to the type of medication used, which can have an impact on the condition, other factors, including the patient's underlying condition, can have an impact on the outcome, and the authors did not consider factors other than medication use.
Response: We appreciate the reviewer's comment on this matter. It is essential to acknowledge that factors such as race and blood eosinophil counts may influence the long-term benefits of macrolide therapy in COPD patients. Nevertheless, it is important to note that there is limited data available from RCTs to support a detailed subgroup analysis. To address this, we have incorporated the following information into the limitation section: "Fourthly, it is worth noting that other variables, such race and blood eosinophil counts, may potentially affect the efficacy of macrolide therapy in long-term use for COPD patients. However, there is a scarcity of data from RCTs that can robustly support a comprehensive subgroup analysis."(Line 395-398)
Comment 4. The formatting of references is not consistent.
We appreciate the reviewer's comment on this matter. We formatted the references using Endnote software and made necessary modifications to the reference content.